# Glucomannan in *Dendrobium catenatum*: Bioactivities, Biosynthesis and Perspective

**DOI:** 10.3390/genes13111957

**Published:** 2022-10-27

**Authors:** Luyan Qi, Yan Shi, Cong Li, Jingjing Liu, Sun-Li Chong, Kean-Jin Lim, Jinping Si, Zhigang Han, Donghong Chen

**Affiliations:** 1State Key Laboratory of Subtropical Silviculture, Zhejiang A&F University, Lin’an, Hangzhou 311300, China; 2National Innovation Alliance of *Dendrobium catenatum* Industry, Engineering Technology Research Center of *Dendrobium catenatum* of National Forestry and Grassland Administration, Hangzhou 311300, China

**Keywords:** *Dendrobium catenatum*, glucomannan, biosynthetic pathway, structure, bioactivity, dietary fiber, hidden hunger, biofortification

## Abstract

*Dendrobium catenatum* is a classical and precious dual-use plant for both medicine and food in China. It was first recorded in *Shen Nong’s Herbal Classic*, and has the traditional functions of nourishing yin, antipyresis, tonifying the stomach, and promoting fluid production. The stem is its medicinal part and is rich in active polysaccharide glucomannan. As an excellent dietary fiber, glucomannan has been experimentally confirmed to be involved in anti-cancer, enhancing immunity, lowering blood sugar and blood lipids, etc. Here, the status quo of the *D. catenatum* industry, the structure, bioactivities, biosynthesis pathway and key genes of glucomannan are systematically described to provide a crucial foundation and theoretical basis for understanding the value of *D. catenatum* and the potential application of glucomannan in crop biofortification.

## 1. Introduction

*Dendrobium catenatum* (also named *D. officinale*) is a rare and precious Traditional Chinese Medicinal (TCM) plant, and it was first recorded in the earliest works of Chinese Medicine, *Shen Nong’s Herbal Classic*, written in the Eastern Han Dynasty nearly 2300 years ago. The main medicinal part of *D. catenatum* is the stem, which has the effects of nourishing *yin*, antipyresis, tonifying the stomach, and promoting fluid [1]. Wild *D. catenatum* is on the verge of extinction, and it was listed as a third-class protected species in the *Regulations on the Protection and Management of Wild Medicinal Resources* issued by the State Council of China in 1987 and as an endangered plant by the *China Plant Red Data Book* in 1992. Subsequently, *D. catenatum* was listed as critically endangered on the International Union for Conservation of Nature (IUCN) Red list (Critically Endangered A4c ver 3.1), and as a second-class national protected endangered plant on the *List of Rare and Endangered Plants of the People’s Republic of China* in 2009 [2]. In the 2010 edition of the *Pharmacopoeia of the People’s Republic of China*, *D. catenatum* (*D. officinale*) was separated from the “*Dendrobium*” item to form a single item. In 2018, *D. catenatum* was listed as the top one of the new “Zhe-Ba-Wei” Chinese Medicinal Materials by Zhejiang province. In 2019, *D. catenatum* was included in the catalogue of pilot work of food and drug material management by the National Health Commission of the People’s Republic of China, which indicates that *D. catenatum* is officially recognized as an edible material. In 2021, as a second-level protected plant, *D. catenatum* was further included in the new version of the *List of National Key Protected Wild Plants* jointly issued by the National Forestry and Grassland Administration and the Ministry of Agriculture and Rural Affairs of the People’s Republic of China.

As a traditional dual-purpose plant for both food and medicine, *D. catenatum* research and industry have experienced three developmental stages (I~III). (I) The initiation stage. Since the 1970s, preliminary studies on the improvement of both seedling propagation and artificial cultivation technologies have been carried out. *D. catenatum* raw materials were obtained from wild resources by unrecoverable over-exploitation. (II) The blooming stage. In the beginning of the 2000s, with breakthroughs in key technologies for seed production, tissue culture, and planting substrates of *D. catenatum*, the artificial-sheltered cultivation mode (Figure 1A) with high yield was dramatically developed, resulting in a sharp rise of a ten-billion-level industry [3]. (III) The plateau stage. From the 2020s onward, simulated wild cultivation mode with *D. catenatum* directly planted on the rock and trunk (Figure 1B,C) is popular due to its high TCM quality, contributing to breaking through the bottleneck and further transformation and upgradation of the *D. catenatum* industry. Therefore, *D. catenatum* has progressed from its natural wild state (endangered) to facility cultivation (high-yield), then to imitation wild cultivation (high-quality), leading to species conservation and industrial rise with combined ecological and economic benefits.

Modern studies have shown that the main active components of *D. catenatum* are polysaccharides, alkaloids, phenols, terpenes, and flavonoids [4]. Among them, polysaccharide is closely related to its pharmacological activity and is an important index for the evaluation of *D. catenatum’s quality*. The main active polysaccharide in *D. catenatum* stem is glucomannan, which has antioxidant and mild immunostimulatory activity to protect macrophages from hydrogen peroxide (H_2_O_2_)-induced oxidative damage [5]. Glucomannan, as an excellent soluble dietary fiber, has the virtues of regulating the intestines and stomach, improving immunity, lowering blood sugar and blood fat, fighting cancer, and losing weight [6,7]. In addition, mannose as the main glucomannan component has been identified to inhibit cancer cell growth by interfering with cellular glucose metabolism [8].

Hidden hunger refers to malnutrition due to nutritional imbalance in the human body, even having taken in enough energy. More than two billion people worldwide have suffered from hidden hunger, which is highly correlated with the status that three staple cereals (rice, maize, and wheat) provide 60% of the world’s food intake [9]. Recently, rediscovery and utilization of a nutrient-rich variety of “forgotten crops” in Asia offers a viable and promising approach to eliminate hidden hunger [10]. Glucomannan belongs to soluble hemicellulose and is an excellent source of soluble dietary fiber as the seventh essential nutrient in the human body. Therefore, incorporating *D.*
*catenatum* as a rich source of glucomannan into the daily diet contributes to promoting nutritional diversification and alleviating hidden hunger.

## 2. Feature and Structure of Glucomannan

Unlike other soluble fibers, glucomannan is characterized by high viscosity [11]. As a hemicellulose polysaccharide, glucomannan is ubiquitous in the plant cell wall. Moreover, glucomannan exists as energy storage substance in Araceae, Liliaceae, and Iridaceae and Orchidaceae [12]. It was reported that glucomannan contributes to plant tolerance to the lack of water as a compatible solute and the succulence in *Aloe vera* [13]. Storage glucomannans in distinct species display different structures. The mannose:glucose (Man:Glc)ratio is 1.6:1.0 in glucomannan from *Amorphophallus konjac* and 3.0:1.0 in glucomannan from *Orchis mascula* [14,15]. The Man:Glc ratios of three glucomannans (ASP-4N, ASP-6N and ASP-8N) in *Aloe* leaves are 19.13:1, 8.97:1, and 2.96:1, respectively [16]. In addition, glucomannan can also be obtained from microorganisms, such as the cell walls of bacteria, yeast, or fungus [17,18]. Natural glucomannan is composed of D-glucose and D-mannose linked by a β-1,4-glucopyranoside bond to form polymer heteropolysaccharides in a certain molar ratio [19]. Moreover, the C3 position of D-mannose on the main chain can be linked to polysaccharides in the form of β-1,3-glycosidic bond or β-1,6-glycosidic bond [20].

Glucomannan is the main polysaccharide active component in *D. catenatum* (Figure 2, Table 1), and its derivatives also contain other monosaccharides, such as galacturonic acid, glucuronic acid, and galactose [21]. The contents of mannose and glucose in total polysaccharides from *D. catenatum* are 120.60 mg·g^−1^ and 71.23 mg·g^−1^, respectively [22]. Several glucomannans from *D. catenatum* have been reported to exhibit different Man:Glc molar ratios [5,23,24], and contain abundant O-acetyl groups [25,26,27]. Acetylation can increase the activity and health benefits of glucomannan [28].

## 3. Applications of Glucomannan

### 3.1. Medical Applications

*D. catenatum* has multiple traditional functions and has been developed into various medicines, such as Mailuoning injection (Jinling Pharmaceutical Co., Nanjing, China), Tongsaimai tablet (Jiangsu Kangyuan Sunshine Pharmaceutical Co., Nanjing, China), and Dendrobium nightlight pill (Tong Ren Tang, Beijing, China). Modern medicine has shown that *D. catenatum* polysaccharides have diverse bioactivities, including immunomodulatory, anti-tumor, gastro-protective, hypoglycemic, anti-inflammatory, hepatoprotective, and vasodilating effects [37]. In this review, major emphasis will be placed on the gastrointestinal protection, immunomodulatory, anti-cancer, hypoglycemic, and hypolipidemic functions of *D. catenatum* glucomannan (Figure 3).

#### 3.1.1. Intestinal Health Improvement

It has been reported that glucomannan can modulate mouse cecal and fecal microbiota with favorable prebiotic effects [38]. The unique properties of glucomannan hydrolysate make it valuable as a prebiotic in a wide range of food, feed, and pharmaceutical products [39]. Prebiotics promote specific changes in the gastrointestinal microbiota [40], promote the growth of probiotics such as Lactobacillus and Bifidobacterium, inhibit the proliferation of harmful bacteria, reduce inflammation, improve the integrity of the intestinal mucosa, promote nutrient absorption [41], and control the blood glucose level of patients with type 2 diabetes [42]. Glucomannan selectively stimulates the production of beneficial gut microflora such as probiotics and benefits the treatment of functional gastrointestinal disorders related to abdominal pain in mice [43,44,45]. The combined laxative of glucomannan-probiotic promotes defecation in constipated rats [46]. Mannose-oligosaccharides, the oxidative degradation product of glucomannan, are potential prebiotics, which can affect the growth and species abundance of fecal microbiota and regulate the balance of intestinal flora [47,48]. For instance, *D. catenatum* polysaccharide DOP can restore the diversity of intestinal flora and regulate the abundance of intestinal flora by inhibiting the overexpression of pro-inflammatory cytokines (TNF-α, IL-6, and IL-1β), restoring the level of short-chain fatty acids (SCFAs), activating G-protein-coupled receptors (GPRs), and regulating the intestinal flora to alleviate the symptoms of colitis in mice [49].

Glucomannan also plays a direct role in protecting the intestinal epithelium. *D. catenatum* polysaccharide DOP is not easily digested and absorbed in the human body but is degraded into SCFAs by the gut microbiota in the large intestine, therefore improving intestinal health [22,50]. In addition, *D. catenatum* glucomannan can reduce intestinal epithelial injury and regulate intestinal mucosal immunity by keeping a balanced ratio of pro- and anti-inflammatory cytokines and regulating the expressions of toll-like receptors (i.e., TLR-2, TLR-4, TLR-6, and TLR-9) important for recognizing pathogen-associated molecular patterns derived from various microbes in mice [51].

#### 3.1.2. Immunomodulatory Activity

Glucomannan, one of the natural bioactive ingredients, can be used as an ideal immunomodulatory agent. Glucomannan can reduce brain inflammation, improve hippocampal neuron damage, maintain hippocampal cognitive function, and play an anti-epileptic role in epileptic rats [52]. *D. catenatum* polysaccharide DOPW3-B can improve the intestinal mucosal immune activity by increasing at Peyer’s patches the levels of interferon-γ (IFN-γ) and interleukin-4 (IL-4), two key effector cytokines for the differentiation of T helper types 1 and 2 with a positive effect on mesenteric lymph nodes [31,53]. Glucomannan can increase the expression of several cytokines that are important for immune homeostasis (e.g., TNF-α, IL-β and IL-10) [54]. Meanwhile, glucomannan plays a pivotal role in regulating the activation and proliferation of macrophages [55], and promoting phagocytosis [29]. Finally, the pretreatment of *D. officinale* with organic solvents enhances the immunostimulatory activity of polysaccharides and affects the mannose/glucose ratio of polysaccharides, which plays an important role in immunostimulation [56].

#### 3.1.3. Anti-Cancer Activity

In a zebrafish xenograft model, *D. catenatum* polysaccharide DopW-1 inhibits the proliferation of HT-29 cells by the apoptosis pathway and has an anti-tumor effect on colorectal cancer [57]. It is known that hyper-activation of the phosphatidylinositol 3 kinase/protein kinase B (PI3K/AKT) signaling pathway in human cancers can promote the proliferation and survival of tumor cells [58]. Glucomannan can block the PI3K/AKT pathway, thereby promoting the apoptotic rate and reducing the proliferation ability of tumor cells [59]. In parallel, glucomannan can also inhibit the expression of major chemokine receptors, chemokine receptor 4 (*CXCR4*) and CC chemokine receptor 7 (*CCR7*), found in a wide range of tumor cells, thus reducing the dissemination ability of tumor cells [60]. In addition to directly interfering with tumor cells, glucomannan also acts in an indirect manner. The *D. catenatum* polysaccharide DOPa-3 significantly inhibits the formation and growth of colon tumors and alleviates colon injuries [61]. In rats, DOPA-4 effectively inhibits precancerous lesions of gastric cancer induced by 150μg/mL MNNG [62]. *D. catenatum* polysaccharides can reduce oxidative stress level, inhibit stress-induced activation of adenosine monophosphate-activated protein kinase (AMPK)/UNC-51 like autophagy activating kinase 1 (ULK1) pathway and the expression of light chain 3 (LC3) I and LC3 II proteins, reduce Beclin1 expression, and then reduce hypoxia/reoxygenation induced astrocyte autophagy, reduce human astrocyte apoptosis, and promote astrocyte survival [63].

#### 3.1.4. Hypoglycemic Activity

Glucomannan is suitable as a dietary fiber supplement for the treatment of being overweight, hyperlipidemia, and diabetes. *D. catenatum* glucomannan DOP can alleviate hyperglycemia in high fat diet (HFD)/streptozocin (STZ)-induced diabetic mice through promoting the synthesis of liver glycogen and inhibiting the degradation of liver glycogen [64].

DOP treatment can reduce the level of bile acid in diabetic rats, reduce the binding of bile acid to the nuclear receptor FXR or the membrane receptor TGR5, increase the level of glucagon-like peptide-1 (GLP-1), and improve glucose and lipid metabolism and insulin sensitivity [65]. DOP promotes glycogen synthesis by regulating the expression of glycogen synthase kinase 3β (GSK-3β) and glycogen synthase (GS) in the liver or glucose transporter 4 (GLUT4) in muscle. Glucose levels are reduced by regulating the activity of glucose metabolism enzymes in the liver, including pyruvate kinase (PK), hexokinase (HK), and phosphoenolpyruvate carboxykinase (PEPCK) [66]. On the other hand, DOP can delay diabetic cataract by decreasing the level of serum malondialdehyde (MDA), increasing the activity of superoxide dismutase (SOD) and enhancing its antioxidant capacity [49]. Glucomannan AABP-2B, isolated and purified from *Anemarrhena asphodeloides*, demonstrates its hypoglycemic effect by inhibiting α-glucosidase activity and activating irS-1/PI3K/Akt signaling pathway in insulin-resistant cells [67]. Effects of glucomannan on insulin sensitivity contribute to weight loss, and taking as little as 4 g of glucomannan per day can promote weight loss [68].

### 3.2. Daily Application

#### 3.2.1. Cosmetics

*D. catenatum* extracts are used as cosmetic raw materials to effectively solve the skin problems caused by *yin* deficiency and fire hyperactivity due to their rich polysaccharides, flavonoids, and other nutrients that are absorbed through the skin and have good moisturizing, anti-aging, and anti-wrinkle effects [69]. The aqueous extract of *D. catenatum* can resist drying damage to epidermal cells, increase cell vitality, and improve skin moisture content [70]. *D. catenatum* polysaccharide DSP has strong antioxidant activity, can scavenge DPPH free radicals, and has a better inhibitory effect on lipid peroxidation and oxidative damage of red blood cells [71,72]. Macromolecular polysaccharide as the main moisturizing component in *D. catenatum* is a kind of polyhydroxyl polymer, whose polar groups can form hydrogen bonds with water molecules to bind water. Meanwhile, polysaccharide can form a uniform film on the skin surface to prevent water loss [73]. *Dendrobium huoshanense* polysaccharide has certain hygroscopic and moisturizing properties with a higher moisture retention rate than glycerin, is non-irritating to skin, and serves as a natural moisturizing agent [74]. So far, many related skin care products are popular in the market, such as moistening and skin brightening masks with *D. catenatum* (SENYU, Jinhua, China), *Dendrobium* emulsion (MISS QUEEN, Ningbo, China), and *D. catenatum* skin corset firming lotion (BOTANIERA, Hangzhou, China).

#### 3.2.2. Food and Functional Food

Glucomannan has wide application prospects in the food industry and can be used as a food additive, meal substitute food, and health care products, such as *Tiepifengdou* capsules, oral liquid, decoction pieces, yogurt [75], teabags [76,77], beverages, and noodles [78], and has been applied to a number of patents [79]. Due to its large molecular weight and strong water binding ability, glucomannan has excellent hydrophilicity, gelation, emulsification, film formation, thickening, and other unique functional properties [80]. With alkali treatment, the acetyl group in glucomannan is removed to promote the formation of intramolecular and intermolecular hydrogen bonds and get a gel with excellent stability [81]. Glucomannan is a licensed food additive, which is used as a stabilizing, thickening, and gellating agent [82]. For example, in yogurt and other drinks, glucomannan can be used as a thickening agent to increase flavor and nutrition [82]. In sausages, hams, and other meat products, glucomannan can replace fat, reduce fat content, increase viscosity, and water retention, to improve the texture and flavor of meat [83]. In starch products, the addition of glucomannan affects the paste characteristics, rheological properties, and texture of the starch system [84]. In addition, glucomannan can be used for food preservation, such as glucomannan film, which has good stability and food applicability and can be used for fruit and vegetable coating preservation and flavor microcapsule production [85].

## 4. Glucomannan Biosynthesis Pathway in Plant

In the biosynthesis pathway of glucomannan (Figure 4), the photosynthates of leaves are transported to the *D. catenatum* stem in the form of sucrose, which is then decomposed into glucose (Glc), UDP glucose (UDP-Glc), and fructose (Fru) under the action of sucrose synthase (SUS) and invertase (INV), and then, under the action of hexokinase (HXK) and fructokinase (FRK), Glc-6-P is further catalyzed by phosphoglucomutase (PGM) to produce glucose-1-phosphate (Glc-1-P). At the same time, Fru-6-P is catalyzed by phosphate mannose isomerase (PMI) to produce mannose-6-phosphate (Man-6-P), which is converted to mannose-1-phosphate (Man-1-P) by phosphomannomutase (PMM). GDP mannose pyrophosphorylase (GMP) catalyzes the production of GDP mannose (GDP-Man). Glc-1-P is converted to GDP-glucose (GDP-Glc) by GDP-Glc pyrophosphorylase (GGP) and Glc-1-P is also converted to ADP-glucose (ADP-Glc) by ADP-Glc pyrophosphorylase (AGP). Subsequently, GDP-Man, GDP-Glu, or ADP-Glu are each transported into the Golgi apparatus by specific transporters, and they are used as substrates of cellulose-like synthases A/D (CSLA/D) to synthesize glucomannan. Additionally, GDP-Man is also used for the synthesis of vitamin C/ascorbic acid (AsA) [86]. Glucomannan synthesized in the Golgi matrix might have two roles. On the one hand, it is localized in the cell wall through vesicle transport and functions as a structural polysaccharide [87]. On the other hand, glucomannan also acts as a storage polysaccharide in some plants, such as *D. catenatum*, *Konjac*, and *A. vera*. In *Konjac*, glucomannan was discovered to accumulate in the egg-shaped idioblast within the parenchyma [12].

## 5. Research Progresses of Key Glucomannan Biosynthetic Genes in Plant

The pathway of glucomannan synthesis involves some specific key enzymes: phosphate mannose isomerase (PMI), phosphomannomutase (PMM), GDP-mannose pyrophosphorylase (GMP), and cellulose-like synthase A/D (CSLA/D). PMI, PMM, and GMP can provide precursors for the synthesis of GDP-Man, which serves as not only the glycosyl donor and metabolic intermediate widely existing in various organisms but also the substrate of the biosynthesis of glucomannan and ascorbic acid (AsA) [86]. In *D. catenatum*, the expression levels of *DcPMI*, *DcPMM*, *DcGMP2*, and *DcCSLA4/8/13* in the stem are significantly higher than those in other tissues (Figure 5), indicating their important roles for glucomannan accumulation in the *D. catenatum* stem.

### 5.1. Phosphate Mannose Isomerase (PMI): Fru-6-P ↔ Man-6-P

Phosphate mannose isomerase (PMI) catalyzes the reversible conversion between Fru-6-P to Man-6-P in eukaryotes and prokaryotes and is a key enzyme during GDP-Man production [90], required for the first step of the mannose/L-galactose pathway during AsA biosynthesis in plants [91]. In Arabidopsis, there are two PMI1 isozymes, but PMI1, rather than PMI2, is involved in AsA biosynthesis. PMI1 has constitutive expression in both vegetative and reproductive organs under normal growth conditions, whereas PMI2 has no expression in any organs under light. Continuous light can induce PMI1 expression and an increased AsA level in leaves, whereas long-term darkness can induce PMI2 expression and decrease the AsA level. The diurnal expression pattern of PMI1 is in parallel with the total PMI activity and the AsA content in leaves. Moreover, knockdown of PMI1 results in a substantial decrease in the total AsA content of leaves, whereas knockout of PMI2 does not affect the total AsA levels in leaves [92]. In addition, AsA plays an essential role in scavenging reactive oxygen species (ROS) produced during cell metabolism and stress, and increase in the AsA level contributes to an improvement in abiotic stress tolerance [93]. Overexpression of *BcPMI2* from non-heading Chinese cabbage (*Brassica campestris* ssp. *chinensis* Makino) improves AsA content and tolerance to oxidative and salt stress under NaCl or H_2_O_2_ treatment, although transgenic tobacco has lower contents of AsA and soluble sugar than WT under normal conditions [94].

To date, PMI-like genes are mainly used in the mannose selection-based plant transformation system (plant PMI/Man system), which is non-antibiotic and environmentally friendly [95]. Most plant cells are sensitive to mannose and fail to utilize mannose as a sole carbon source because mannose can be taken up and phosphorylated to Man-6-P by endogenous hexokinase, but Man-6-P is not further utilized in the absence of enough PMI activity, subsequently leading to the accumulation of Man-6-P, inhibition of phosphoglucose isomerase, glycolysis blocking, and cell growth arrest [96]. However, plant cells with high PMI activity through genetic modification can metabolize Man-6-P and enter into glycolysis for ATP production and normal growth [97]. As a new safe positive selectable marker gene, *PMI* has been successfully used in the genetic selection of many plants, such as rice, maize, wheat, sorghum, sugarcane, onion, tomato, potato, cassava, grape, apple, cucumber, sugar beet, papaya, Arabidopsis, cabbage, and so on [98,99,100]. The *Es**cherichia coli PMI* (*EcPMI*) gene was first used in the plant PMI/Man system, which is the most popular one [101]. In 2009, the first genetically modified maize MIR604 via the EcPMI/Man selection system was approved for food and feed use, import, and processing in the European Union (European Food Safety Authority, 2009). *Saccharomyces cerevisiae PMI* (*Sc**PMI*) can also be used as a selectable marker in rice transformation [98]. However, non-plant type PMIs can still raise public safety concerns, and plant type PMIs may offer a superior alternative [102]. Hu et al. first proved that plant *PMI* genes from *Chlorella variabilis* and *Oryza sativa* can also be used as selectable markers to obtain transgenic plants exhibiting an accumulation of PMI transcripts and enhancement of PMI activity [100]. Through the selection system based on the green microalga *Chlorococcum* sp. *PMI (ChlPMI)*, a polycistronic gene cluster containing *crtB*, *HpBHY*, *CrBKT*, and *SlLCYB* is transformed into tomato, resulting in the production of high astaxanthin content [103].

### 5.2. Phosphomannomutase (PMM): Man-6-P ↔ Man-1-P

Phosphomannomutase (PMM) catalyzes the interconversion between Man-6-P and Man-1-P and provides precursors for the synthesis of GDP-Man. GDP-Man is not only used for the synthesis of glucomannan but is also an indispensable intermediate in the AsA biosynthesis pathway. The PMM enzyme isolated from the *cinnamon* seed has the activity of converting D-glucose produced by photosynthesis or glycolysis into a mannose component of plant storage polysaccharide [104]. At present, there is much research on the catalytic role of PMM in the AsA synthesis pathway. In tobacco, decreased expression of PMM via the VIGS technique resulted in a reduction of AsA content in leaves, while overexpression of *PMM* increases AsA content in leaves [105]. Transgenic tobacco plants overexpressing the acerola (*Malpighia glabra*) *PMM* gene show around a 2-fold increase in AsA content compared with WT, which correlates with the level of PMM transcripts and the corresponding enzymatic activities [106]. Overexpression of rice PMM in transgenic rice increased AsA content in seeds by 25–50% [107]. Overexpressing *D. catenatum PMM* gene in Arabidopsis, resulting in a significant increase in the contents of both AsA and polysaccharide [108].

### 5.3. GDP-Mannose Pyrophosphorylase (GMP): Man-1-P ↔ GDP-Man

GDP-mannose pyrophosphorylase (GMP) (also named Vitamin C Defective1, VTC1) catalyzes the conversion of Man-1-P and GTP to GDP-Man and pyrophosphate. GMP plays an important role in maintaining the AsA level and redox balance in plants. Compared with WT, the Arabidopsis *vtc1-1* mutant accumulated only 25% of leaf AsA content, but had no effect on the AsA redox state [109]. Arabidopsis KONJAC1 (KJC1) and KJC2 interact with VTC1 to stimulate GMP activity to affect the accumulation of AsA and glucomannan, and VTC1 mutants cause severe dwarfism [110]. The acerola plant has very high AsA levels, consistent with that of the *M. glabra MgGMP* gene promoter, which has higher activity than 35S and Arabidopsis GMP promoters. Transgenic tobacco plants containing the *MgGMP* gene and its original promoter, which displayed about 2-fold increased levels of AsA [111]. Transgenic tomatoes overexpressing a yeast-derived GMP exhibit up to a 31 and 17-fold increase in GMP activity in leaves and green fruit, respectively. The AsA levels increase by up to 70% in leaves, 50% in green fruit, and 35% in red fruit, especially in photosynthesizing organs [112]. The overexpression of tomato *SlGMP3* increases total AsA levels and tolerance to oxidative stress in tomatoes and high or low temperature stress in tobacco, whereas knockdown of SlGMP3 in tomatoes leads to substantially decreased AsA content and a defective phenotype with lesions and senescence due to failing to instantly detoxify ROS [113,114]. Overexpression of tomato GMP in potatoes can improve the content of AsA and dehydroascorbate (DHA) under low temperature stress and enhance the cold tolerance of potatoes [115]. Transgenic tobacco expressing *Pogonatherum paniceum* GMP has a high germination rate and high AsA content under drought and salt stress, and has strong salt tolerance and drought resistance [116]. Overexpression of *D. catenatum* GMP in Arabidopsis results in increased mannose content in water-soluble polysaccharides and enhances salt stress tolerance [117], consistent with salt hypersensitivity in the Arabidopsis *vtc-1* mutant [118].

In addition, the GMPase activity level can regulate the sensitivity of Arabidopsis to ammonium [119]. The *vtc1-1* mutant exhibits stunted root growth with an elevated of NH_4_^+^ efflux at the elongation zone and inhibition of cell elongation in the presence of NH_4_^+^ [120], but this NH_4_^+^-hypersensitive phenotype in mutant is independent of AsA-deficiency. In fact, the *GMP* gene disruption can also cause N-glycosylation disorder of proteins associated with the destruction of hormone homeostasis and the increase of nitric oxide content under high NH_4_^+^ conditions, consequently leading to conditional sensitivity to ammonium ions [121]. However, impaired GDP-mannose biosynthesis and defective N-glycosylation are required for but are not the primary causes of conditional NH_4_^+^ sensitivity in *vtc1-1*, whereas pH alterations in the presence of NH_4_^+^ associated with lost N for assimilation and alkalinization of the cytosol account for the drastic root growth defect in WT and *vtc1-1* [122].

### 5.4. Cellulose-like Synthase A/D (CSLA/D): GDP-Man + UDP/GDP-Glu → Glucomannan

The cellulose synthase gene superfamily consists of the CesA family and 10 Csl families: CslA~CslH, CslJ, and CslM [123,124]. The 30 Arabidopsis CSL proteins is clustered into six families: CslA, CslB, CslC, CslD, CslE, and CslG [125], and the 33 rice CSL proteins are also divided into six families: CslA, CslC, CslD, CslE, CslF, and CslH [126], but only the CslA, CslC, CslD, and CslE families are shared by both. CSL proteins are usually located in the Golgi apparatus, which mediates the synthesis of hemicellulose and then transports it to the cell wall [127]. The CSLA/D families are mainly responsible for the skeleton synthesis of mannan and glucomannan [128,129]. So far, nine *CSLA* and six *CSLD* genes have been identified in Arabidopsis, while 10 *CSLA* and four *CSLD* genes have been identified in rice [130,131], and 13 *CSLA* [132] and eight *CSLD* members have been identified in *D. catenatum* [21].

Heterologous expression assays showed CSLA proteins from a variety of species catalyze the biosynthesis of the β-1,4-mannan or glucomannan backbone in vitro, such as CtManS in guar (*Cyamopsis tetragonoloba*) [133], AtCSLA2/7/9 in Arabidopsis [128], OsCSLA1 in rice [134], PtCslA1 in *Populus trichocarpa* [135], AkCSLA3 in *A. konjac* [136], and TfManS in fenugreek (*Trigonella foenum-graecum*) [137]. In Arabidopsis, the *csla9* mutant exhibits significantly decreased glucomannan, and the *csla2 csla3 csla9* triple mutant is absent of detectable glucomannan in inflorescence stems, although these mutants have no alteration in stem development or strength [138]. The *clsa7* mutant is embryo lethal due to defective embryogenesis with evidently delayed development, abnormal cell patterning, and an arrested globular stage, associated with reduced nuclei proliferation and failed cellularization in the endosperm [139]. Overexpression of AtCSLA2, AtCSLA7, and AtCSLA9 leads to elevated glucomannan content in stems; AtCSLA9 overexpression can rescue the embryo lethality of *csla7*, indicating their functional redundancy [138]. Therefore, AtCSLA2, AtCSLA3, and AtCSLA9 are necessary for glucomannan synthesis in the stem, while AtCSLA7 is necessary for glucomannan synthesis in the embryo [138]. Heterologous expression of the O-fucosyltransferase family member AtMSR1 can enhance the glucomannan synthesis capability of AtCSLA2 and AkCSLA3, possibly via affecting enzymatic activity by protein glycosylation [140,141]. The mannosyl level in stem glucomannans is decreased by around 40% in Arabidopsis *msr1* single mutant and by more than 50% in *msr1 msr2* double mutant [142]. Transcription factors ANAC041, bZIP1, and MYB46 can directly bind the promoter of AtCLSA9 and regulate its expression [143]. Furthermore, overexpression of MYB46 leads to a significant increase in mannan content. In *D. catenatum*, overexpression of DoCSLA6 in *Arabidopsis* promotes mannan biosynthesis [144].

CSLD family members are required for the growth of stem and root, tip growth of root hair and pollen tube, and female gametophyte development and fertility. In Arabidopsis, the *csld2* and *csld3* mutants exhibit root hair bursts [145,146,147,148]; the *csld5* mutant has reduced stem growth [149]; the *csld1* and *csld4* mutants are defective in male transmission and pollen tube production; the *csld2 csld5*, *csld3*, and *csld5* and *csld2*, *csld3*, and *csld5* mutants display dwarfism and severely reduced viability [129]; and *csld2* and *csld3* show synergid cell degeneration during megagametogenesis and reduced pollen tube penetration during fertilization [150]. In rice, the *oscsld4* (*nd1*) mutant is dwarfed and its culm possesses a decreased content of xylan and cellulose but an increased amount of homogalacturonan, whereas disruption of the Arabidopsis *AtCSLD5* gene results in decreased xylan and homogalacturonan synthase activities in *Arabidopsis* [151]. It seems that CSLD proteins are not only limited in (gluco)mannan biosynthesis. At first, researchers tended to believe that CSLD proteins have major functions in mannan synthesis because microsomes isolated from tobacco (*Nicotiana benthamiana*) leaves heterologously expressing AtCSLD5 or co-expressing AtCSLD2/3 have elevated mannan synthase activity, specifically using GDP-Man as an activated nucleotide-sugar donor [152]. However, genetic rescue assays with CSLD-CESA chimeric proteins and in vitro biochemical reconstitution suggested that CSLD3 prefers to function as a UDP-Glu-dependent β-1,4-glucan synthase and forms a protein complex displaying a similar ultrastructural feature to the CESA6-forming complex [153,154]. Overexpression of cotton *GhCSLD3* in Arabidopsis enhances primary cell wall synthesis and restores the defects of the *atceas6* mutant, including significantly reduced cellulose content, a defected cell wall, and a lower dry mass, indicating that GhCSLD3 and AtCESA6 may play a similar role in cellulose or cellulose-like polysaccharide synthesis [155]. Furthermore, the CSLD family is the most similar of the *CSL* gene families to the CESA family at the amino acid level [125] and, together with the CSLF family (accounting for mixed-linkage glucan synthesis), displays the closest relationship with the CESA family at the phylogeny level [156]. Therefore, the CSLD family might function as both a (gluco)mannan synthase and a β-1,4-glucan synthase.

## 6. Summary and Perspective

Food production concerns a new hot topic from meeting the need of “eating fully” to “eating healthily” [7,157]. In modern life, dietary structure and habits have changed, and the source of staple food is mostly limited to several starch crops, including rice, wheat, maize, and potatoes, leading to nutrition imbalance and hidden hunger [158]. Furthermore, severe malnutrition will hinder human growth and development, accompanied by mental disorders and diseases [159]. More than 7% of chronic diseases in contemporary society are caused by hidden hunger. The FAO proposed *Zero Hunger* as the 2nd Sustainable Development Goal to eliminate all forms of hunger by 2030, and biodiversity for food and agriculture is indispensable to achieve the *Zero Hunger* program [160]. Dietary diversity and a higher intake of dietary fiber can reduce the risk of chronic diseases such as diabetes, obesity, and cardiovascular disease. Intriguingly, *D. catenatum* as a traditional dual-purpose plant for both food and medicine in China, is able to supple glucomannan as an excellent dietary fiber and has great potential to fight hidden hunger.

The efficiency of *D. catenatum* is largely dependent on the active polysaccharide glucomannan, which has multiple medicinal effects such as anti-tumor, lowering blood lipids, and preventing diabetes. To date, most of the studies on the glucomannan biosynthetic pathway have focused on its role as a cell wall structural component. PMI, PMM, and GMP sequentially catalyze the production of GDP-mannose, thus providing precursors for the synthesis of glucomannan, and CSLA/D is required for the synthesis of the glucomannan skeleton. However, little is known about the subcellular localization, transportation, and regulation of storage glucomannan, which may be clarified by the comprehensive application of newly developed technologies including immunolocalization, high throughput omics, and gene editing, consequently contributing to creating a glucomannan biofortified crop through synthetic biology (Figure 6). For instance, a stack of four identified committed genes, including *manC* and *manB* from *Escherichia coli BL21(DE3), manA and pgi* from *Bacillus subtilis*, are introduced into food-grade *B. subtilis* to obtain mannan [161]. Four anthocyanin synthetic genes, *sZmPSY1*, *sPaCrtI*, *sCrBKT*, and *sHpBHY*, are introduced into rice endosperm to produce astaxanthin-rich rice with high antioxidant activity [162]. Therefore, it is promising that glucomannan biofortified crops in the future will contribute to reducing hidden hunger and chronic diseases and promoting human health.

## Figures and Tables

**Figure 1 genes-13-01957-f001:**
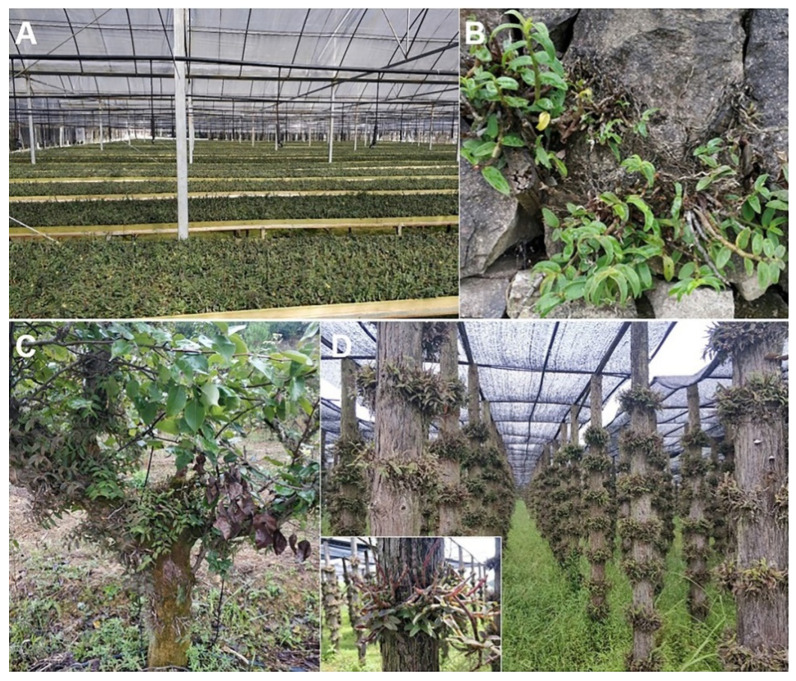
Epiphytic cultivation model of *D. catenatum*. (**A**). Facility-aided cultivation; (**B**) Rock-dependent eco-cultivation; (**C**) Trunk-dependent eco-cultivation; (**D**) Stereo cultivation.

**Figure 2 genes-13-01957-f002:**
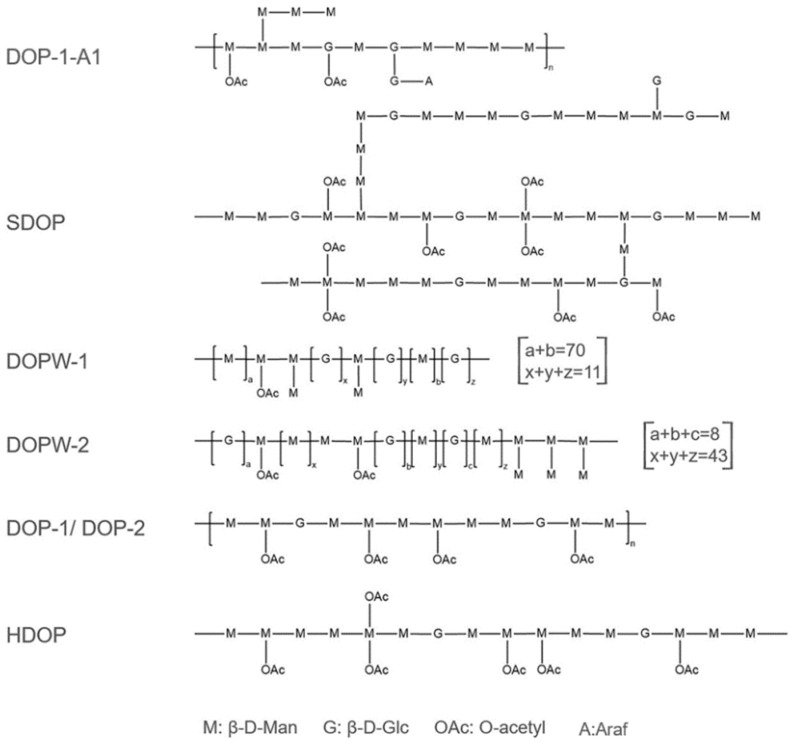
Compositions of different glucomannans in *D. catenatum* ([27,29,30]).

**Figure 3 genes-13-01957-f003:**
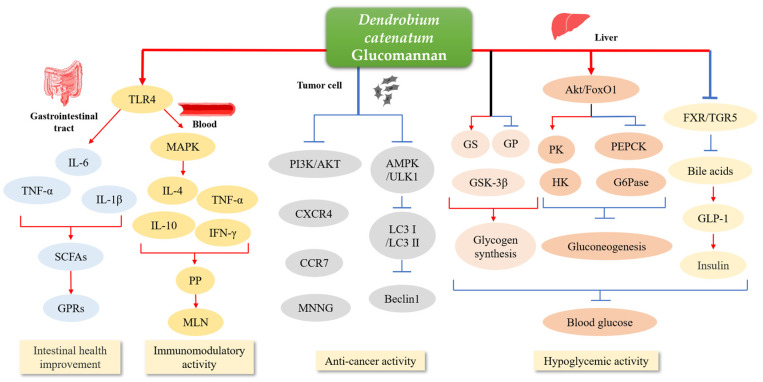
The effects of glucomannan from *D. catenatum* stem. DOP, *Dendrobium officinale* polysaccharide; TLR4, toll-like receptor4; IL-4, Interleukin-4; IL-6, Interleukin-6; IL-10, Interleukin-10; IL-1β, Interleukin-1β; TNF, tumor necrosis factor; PI3K/AKT, phosphatidylinositol 3 kinase/protein kinase B; CXCR4, chemokine receptor4; CCR7, CC chemokine receptor 7 CC chemokine receptor 7; AMPK, adenosine monophosphate-activated protein kinase; ULK-1, UNC-51 like autophagy activating kinase 1; LC3, light chain 3; SCFAs, short-chain fatty acids; GPRs, g-protein-coupled receptors; MAPK, mitogen-activated protein kinase; PP, Peyer’s patche; MLN, mesenteric lymph node; MNNG, 1-methyl-2-nitro-1-nitroguanidine; GS, glycogen synthase; GP, glycogen phosphorylase; GSK-3β, glycogen synthase kinase 3β; PK, pyruvate kinase; HK, hexokinase; PEPCK, phosphoenolpyruvate carboxykinase; G6Pase, glucose-6-phosphatase; GLP-1, glucagon-like peptide-1.

**Figure 4 genes-13-01957-f004:**
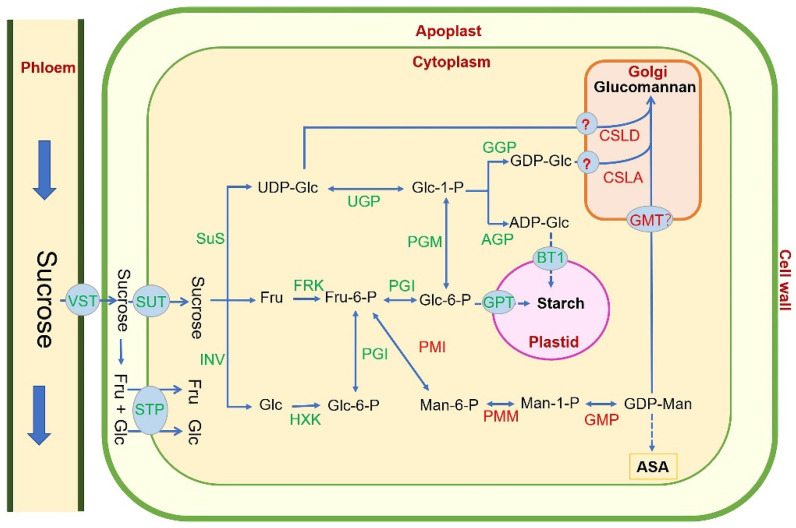
Putative glucomannan biosynthetic pathway in *D. catenatum* (modified from [41,44,45,88] Glc, glucose; Fru, fructose; Fru-6-P, fructose-6-phosphate; Glc-1-P, glucosophosphate-1-P; Glc-6-P, glucosophosphate-6-P; Man, mannose; Man-6-P, mannose-6-phosphate; Man-1-P, mannose-1-phosphate; AsA, ascorbic acid; SUT, sucrose transporter; STP, sugar transporter proteins; SuS, sucrose synthase; INV, invertase; FRK, fructokinase; HXK, hexokinase; PGI, phosphoglucose isomerase; UGP, UDP-Glc pyrophosphorylase; AGP, ADP-Glc pyrophosphorylase; GGP, GDP-Glc pyrophosphorylase; PGM, phosphoglucomutase; GPT, glucose-6-phosphate transporter; BT1, brittle-1 protein, an ADP-Glc transporter; PMI, phosphate mannose isomerase; PMM, phosphomannomutase; GMP, GDP-mannose pyrophosphorylase; GMT, GDP-mannose transporter; CSLA, cellulose-like synthase A; CSLD, cellulose-like synthase D.

**Figure 5 genes-13-01957-f005:**
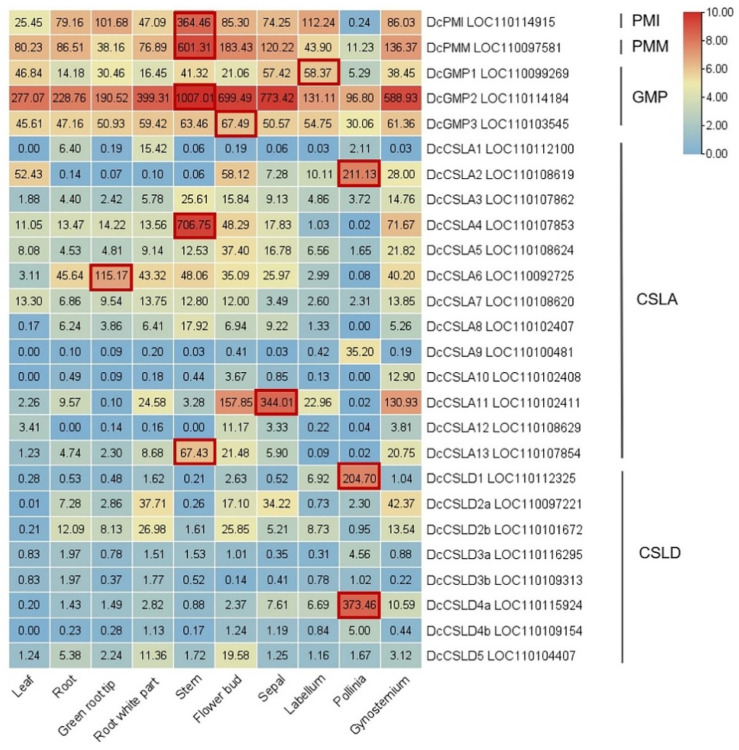
Expression profile of glucomannan pathway genes in tissues and organs of *D. catenatum*. The raw RNA-seq reads of different tissues and organs in *D. catenatum* are derived from the NCBI database (www.ncbi.nlm.nih.gov, accessed on 16 September 2022), including the leaf (SRR4431601), the root (SRR5722140), the green root tip (SRR4431599), the white part of the root (SRR4431598), the stem (SRR4431600), the flower bud (SRR4431603), the sepal (SRR4431597), the labellum (SRR4431602), the pollinia (SRR5722145), and the gynostemium (SRR4431596). A heatmap has been generated via TBtools software [89]. The color scale represents log2 of FPKM expression values; green and red indicate a low and high level of gene expression, respectively.

**Figure 6 genes-13-01957-f006:**
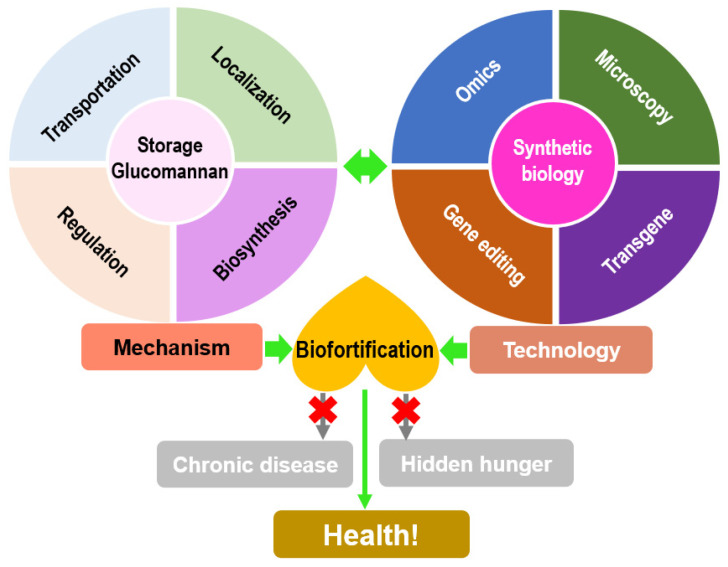
Biofortification strategy of glucomannan based on the in-depth dissection of molecular mechanisms and combined application of new-developed technologies.

**Table 1 genes-13-01957-t001:** Polysaccharides isolated from *D. catenatum*: monosaccharide compositions, molecular weights, structure unit/backbone chain, and bioactivities.

Name	Molecular Weights (Mw, kDa)	Monosaccharide Compositions	Bioactivities	References
DOP2	4699	Man:Glc = 7.64:1.00	Unknown	[23]
DOP3	5480	Man:Glc = 4.50:1.00	Unknown	[23]
DOP4	5408	Man:Glc = 3.57:1.00	Unknown	[23]
DOP-1-A1	130	Man:Glc = 40.2:8.4	Unknown	[25]
SDOP	1660	Man:Glc = 4.9:1.0	Unknown	[26]
DOPW-1	389.98	Man:Glc = 10.75:1.00	Unknown	[29]
DOPW-2	374.11	Man:Glc = 8.82:1.00	Unknown	[29]
DOP-1	389.98	Man:Glc = 5.18:1.00	Immunomodulatory activity	[27]
DOP-2	374.11	Man:Glc = 4.78:1.00	Immunomodulatory activity	[27]
DOP-W3-b	15.43	Man:Glc = 4.5:1.0	Immunomodulatory activity	[31]
DOP-I-1	730	Man:Glc = 5.8:1.0	Immunomodulatory activity	[32]
DOPa	810	Man:Glc = 5.6:1.0	Immunomodulatory activity	[32]
DOPb	670	Man:Glc = 5.9:1.0	Immunomodulatory activity	[32]
DOPA-1	394	Man:Glc = 5.8:1.0	Immunomodulatory activity	[5]
DOPA-2	362	Man:Glc = 4.5:1.0	Immunomodulatory activity	[5]
DWDOP1	1341	Man: Glc = 6.79:1.00	Unknown	[33]
FWDOP1	1415	Man: Glc = 7.46:1.00	Anti-tumor activity	[33]
DOPA-1	229	Man:Glc:Gal = 1.00:0.42:0.27	Anti-tumor activity	[34]
DOP1-DES	298	Man:Glc = 2.2:1.0	Unknown	[35]
DOP2-DES	30	Man:Glc = 3.7:1.0	Unknown	[35]
LDOP-1	91.8	Man:Gal:Glc:Gal:Ara = 2.0:1.7:1.3:1.6:0.7	Anti-inflammatory activity	[30]
DLP-1	1380	Man:Glc = 71.69:22.89	Immunomodulatory activity	
DCP	221	Man:Glc:Gal = 69.5:30.2:0.3	Immunomodulatory activity	[36]

DOP/DOPA/DWDOP/FWDOP/LDOP, *Dendrobium officinale* polysaccharide; SDOP, single *Dendrobium officinale* polysaccharide; DOPW, water-soluble *Dendrobium officinale* polysaccharide fraction; DCP, *D. catenatum* polysaccharides; Man:Glc, mannose:glucose.

## Data Availability

The data used to support the findings of this study is available.

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
