# Peer review of "Glucomannan in Dendrobium catenatum: Bioactivities, Biosynthesis and Perspective"

_genes, 2022, doi:10.3390/genes13111957_

Round 1

Reviewer 1 Report

The manuscript focused on the glucomannan from Dendrobium catenatum is very interesting and easy to read. I appreciate the clear scheme of glucomannan biosynthesis (Fig. 4). But I suggest inserting some information.

Lines 25 - 45: Authors described the history of the Dendrobium catenatum in China. I miss information about this plant from a global perspective.

Line 245 – 268: In this part/sentences, there is too much information without citations/references. Please, insert them.

Line 285 - 293: The same problems with citations as above mentioned.

The authors described the effects of glucomannan on animals and people but I did not find information/part about the effects of the (galacto)glucomannan on plants.  

I found some grammar mistakes in the whole text, including the Latin names of plants without italics. I suggest reading the manuscript carefully.

Author Response

Lines 25 - 45: Authors described the history of the Dendrobium catenatum in China. I miss information about this plant from a global perspective.

Answer: We only focus on the edible and medicinal history of Dendrobium, mainly in China. Other countries have limited records about it.

Line 245 – 268: In this part/sentences, there is too much information without citations/references. Please, insert them.

Answer: According to another review’s opinion, this paragraph has been deleted, because it's too basic.

Line 285 - 293: The same problems with citations as above mentioned.

Answer: Ok, I have added citations.

The authors described the effects of glucomannan on animals and people but I did not find information/part about the effects of the (galacto)glucomannan on plants.  

Answer: There's limited relevant literature, but I've added. Please see L92.

I found some grammar mistakes in the whole text, including the Latin names of plants without italics. I suggest reading the manuscript carefully.

Answer: Ok, I have revised.

Reviewer 2 Report

This manuscript is about the roles, applications, biosynthesis of glucomannan polysaccharide from Dendrobium catenatum, a traditional Chinese medicinal plant. Authors have covered various applications to health, food, and plants from the glucomannan. Authors have suggested biofortification of glucomannan to crops could be very useful for future to promote health. This paper is good for international readers to know about such a beneficial plant.

Major suggestions:

·       Unfortunately, the manuscript is not well-organized with title – “function, biosynthesis, perspective(s)” – function is interchangeably used as roles in plants or benefits in animal and human or application or genetic or biochemical uses. This must be confusing for the readers.

·       There should be proper subtitles can be given to explain each. For example, biosynthesis is hidden somewhere between the benefits or applications.

·       This paper need to be organized against the titles and subtitles and then cover specific aspects in those subtitles.

·       In many places, it is too technical and descriptive. It may not be necessary to explain the technical aspects or describe fully on the process to link this glucomannan to applications.

·       This manuscript needs to be concise for any reader. It should attract international readers not just Chinese readers. No references for many statements and several statements are hypothetical guesses. This has to be avoided in the review papers as only original research can be reviewed and authors can have an opportunity to suggest based on available evidences.

Other suggestions:

·       How glucomannan supplement nutritional imbalance of rice, maize, wheat diet or hidden hunger? Any reference? Evidence?  

·       Intestinal health improvement: only mouse and rats were mentioned. Only one human study indicated. Other studies stated without the model organism. If there are human studies then should indicate clearly. Otherwise, it is still in research and not proved much in human health.

·       Many studies are not indicating model organism?

·       Clinical application: more emphasis was given to cellular and molecular activity rather than linking the outcomes of those studies to health benefits.

·       No commercial products or medicines were indicated. This means it is still in research only, not in the market. If products are available in the market, then it would great to state them.

·       Focussed on several benefits, rather it would be good to select the best ones and discuss more. Others can be just mentioned or discussed less.

·       Grammar needs to be checked throughout the manuscript

·       Summary: Rather than providing long summary, it can be provided with catchy concluding remarks.

Minor suggestions

·       Abstract

·       Page 1 Line 14 and 29: Clearing heat?

·       Page 1 Line 27: Chinese letters? International readers so convert into English?

·       Line 34 IUCN – expand first time?

·       Line 37 was separated from “Dendrobium” item to form a single standard – not understandable, need to explain more?

·       LINE 41 in according with that some Chinese herbal medicines 41 are widely used as food materials in China traditional food culture – English?

·       Page 2 line 47 three developmental stages. (I) Initiation stage – list all stages here, and then explain one by one?

·       Line 48: Since the 1970’s should be Since 1970’s?

·       Line 56: (What) Bottlenecks (were not mentioned)

·       L58: need to refine the sentence and words in brackets

·       L63: could be “and is an important index for the evaluation of its quality”

·       L74: (REF)?

·       L75: rich variety? Or nutrients-rich variety?

·       L78: need to refine the sentence, “to promote” not “to promoting”

·       L89: “The Man: Glc” should be expanded in first as new readers would not know what it is? No space between Man:Glc.

·       Table 1: All names are abbreviated, need to write what is DOP2, DOP3 etc? and Man:Glc? at the end of the table. Table should be stand alone.

·       Fig 3: Many genes were just abbreviated, only a few were expanded in legend.

·       L160: result was stated as “may enhance” that means study has not yet been proved.

·       L208: “D. catenatum extracts can be used” or D. catenatum extracts is used?

·       L220: “potential to serve as natural moisturizing agent” – this means it has not yet been used.

·       L242: “In cooked meat products, glucomannan can replace fat, reduce fat 242 content, increase consistency and water retention, and improve meat texture” – need more explanation.

·       L245-251: Just a description, not necessary to this paper.

·       L252-264: Descriptive paragraph. Can be avoided.

·       L270 and L272: “still remains mysterious” and “still needs to be further determined” – unproved studies can be avoided.

·       L285-290: descriptive

·       L292: “important roles” - what are those roles?

·       L302 and 303: “PMI” - Title and paragraph should be avoided starting with abbreviations

·       L322: no reference?

·       L329: “As a new safe positive selectable marker gene, PMI has been successfully used in genetic selection of many plants” – need to explain for readers, how this was used as a marker gene?

·       L346-351: descriptive

·       L352: “via” italics

·       L361-362: descriptive

·       L459-461: no reference for such an important statement?

·       L492: “Therefore, it is promising that glucomannan biofortified crop in the future will contribute to reduce hidden hunger and chronic diseases, and promote human health” – if there are not enough benefits directly linked to hidden hunger, chronic diseases and human health then this is too hypothetical to state such glucomannan is going to solve such a huge problems.

·       L111: “various” – various?

Round 2

Reviewer 2 Report

This manuscript has been sufficiently revised and improve for the publication in Genes.